# Development of a Fermented Beverage with *Chlorella vulgaris* Powder on Soybean-Based Fermented Beverage

**DOI:** 10.3390/biom13020245

**Published:** 2023-01-27

**Authors:** Norbert-Istvan Csatlos, Elemer Simon, Bernadette-Emőke Teleky, Katalin Szabo, Zorița Maria Diaconeasa, Dan-Cristian Vodnar, Călina Ciont (Nagy), Oana-Lelia Pop

**Affiliations:** 1Department of Food Science, Faculty of Food Science and Technology, University of Agricultural Science and Veterinary Medicine, 400372 Cluj-Napoca, Romania; 2Institute of Life Sciences, University of Agricultural Sciences and Veterinary Medicine of Cluj-Napoca, 400372 Cluj-Napoca, Romania; 3Molecular Nutrition and Proteomics Lab, CDS3, Life Science Institute, University of Agricultural Science and Veterinary Medicine, 400372 Cluj-Napoca, Romania

**Keywords:** fermentation, soy beverage, *Chlorella vulgaris*, probiotics, digestion

## Abstract

The area of functional beverages made from plant-based or non-dairy milk is one of the fastest-growing sectors in the world. The microalgae *Chlorella vulgaris* is a source of functional ingredients, with a large spectrum of healthy compounds, such as canthaxanthins, astaxanthins, peptides, and oleic acid. The study aimed to investigate the suitability of *C. vulgaris* biomass as a substrate for *Lactobacillus fermentum* and *Lactobacillus rhamnosus* development and fermentation in vegetal soy beverages and to evaluate the fermented product in terms of bacterial viability, antioxidant capacity, and in vitro bio-accessibility. During fermentation, a bacterial concentration of 8.74 log10 CFU/mL was found in the soy beverage with *C. vulgaris* and *L. rhamnosus*, and 8.71 log10 CFU/mL in beverage with *C. vulgaris* and *L. fermentum*. Polyphenol content and dietary antioxidant capacity significantly improved after fermentation soy drinks. On the other hand, through the digestibility of the beverages, the bacterial viability significantly decreased. To comprehend the components responsible for the efficient delivery of bacteria across the gastrointestinal tract, further investigation is required on probiotic encapsulation methods.

## 1. Introduction

One of main challenges of the food industry is to develop improved-quality vegan products with enhanced nutritional, functional, and sensory characteristics, as, currently, 79 million people are following a vegan diet [1]. In this context, the scientific community investigates natural and sustainable ingredients to design innovative foods [2,3,4]. Fermented products have received much attention recently due to their positive influence on gut microbiota [5,6]. Fermented foods are defined as “foods or beverages produced by controlled microbial growth and transformation of food components through enzymatic action” [7,8]. Fermentation has been widely used to increase the bioavailability of nutrients and to reduce anti-nutritional factors in soy [9,10]. Several studies have also confirmed the ability of the fermentation process to degrade the anti-nutritive and allergenic compounds of soybeans [11,12,13,14,15,16]. Fermented soy products have significantly lower levels of anti-nutritional elements such as phytates, trypsin inhibitors, and lectins, than raw soy products [17,18]. In particular, lactic acid bacteria (LAB), mediated fermentation can decrease phytate and trypsin inhibitors and hydrolyze tannic acid [19,20]. Moreover, during the fermentation process, many new compounds are synthesized, such as isoflavones, water-soluble vitamins, and vitamin K2 (menaquinone-7), which play a significant role in human health [21,22,23]. Probiotics have been implemented in a wide range of food products, including dairy products, meat, beverages, cereals, vegetables, fruits, and bread or other bakery products [24]. However, probiotics must be sustained at significant concentrations during the product’s shelf life [25].

Microalgae are a viable source of bioactive compounds for functional food products that can be used as prebiotics for probiotics [26,27,28,29]. Before releasing a new microalgal-based food for dietary purposes, it must be shown that the ingredient is safe. Thus, de Mello-Sampayo et al. [30] evaluated carotenogenic biomass (orange) resulting from induced stresses of *C. vulgaris*. The consumption of *C. vulgaris* did not reveal any indicators of toxicity at dosages surpassing the suggested human carotenoid intake level. These findings point to the potential of microalgae as a source of carotenoids, suggesting it may have health advantages if consumed by humans [30]. Several researchers have investigated the effect of adding microalgae to food products, such as biscuits, bread, pasta, yogurt, cheese, and fermented milk beverages [17,28,31,32,33,34]. For yogurts, Barkallah et al. reported that incorporating 0.25% Spirulina biomass improved fermentation and promoted syneresis and antioxidant activity [35]. Results presented by Mazinani et al., suggested that the addition of *Spirulina platensis* (0.3%, 0.5%, and 0.8%) increased hardness and probiotic viability (*Lactobacillus acidophilus*) in cheese [36], whereas *C. vulgaris* (1%, 2%, and 3%) enhanced firmness and springiness, but decreased cohesiveness and meltability [37]. Moreover, *C. vulgaris* and *Spirulina maxima* biomass (0.5%, 1.0%, and 2.0%) added to fresh spaghetti improved the firmness and color of the product (green and orange), making it more attractive to consumers [38]. On the other hand, microalgae-based proteins could significantly contribute to meeting the population’s need for protein, with several advantages over other protein sources currently in use. Microalgae-based proteins have low land requirements (< 2.5 m^2^/kg protein) compared to animal proteins production, such as pork (47–64 m^2^/kg protein), chicken (42–52 m^2^/kg protein), and beef (144–258 m^2^/kg protein) [34]. To better understand the action of biological compounds from *C. vulgaris* powder, the present study was conducted to investigate the effects of the addition of microalgae on microbiological and biochemical characteristics of probiotic soy beverages for possible use as a functional and lactose-free product.

## 2. Materials and Methods

### 2.1. Materials

The fermentation involved two probiotic strains, *Lactobacillus fermentum* (LMG 6902) and *Lactobacillus rhamnosus* (LMG 25626), which were obtained from BCCM/LMG Bacteria Collection. Dried de Man, Rogosa, and Sharpe (MRS) growth media were acquired from HIMEDIA (Einhausen, Germany).

The commercial soy beverage originated by Dennree (Allemagne, Germany), was purchased from a food store (Cluj-Napoca, Romania). The raw materials of the soy beverage were water and 8% soy from ecologic farming, having 1.5% fat, 0.9% carbohydrates, and 3% proteins. The product was UHT pasteurized. Xylitol, and inactivated dried organic *Chlorella vulgaris* powder were acquired from specialized stores in Cluj-Napoca, Romania. The country of origin this powder was China, having 60% proteins, 12% fibers, 9.5% carbohydrates, and 7.9% fat.

The enzymes used for the simulation of gastrointestinal digestion, pepsin from porcine gastric mucosa (P6887), pancreatin from porcine pancreas (P7545), and bovine bile extract (B8631), were acquired from Sigma-Aldrich (Taufkirchen, Germany). The chemicals for the biological characterization, DPPH (1,1-diphenyl-2-picrylhydrazyl), Trolox, and Folin–Ciocâlteu reagent were also purchased from Sigma-Aldrich. All the materials and chemicals used in the experiment were of analytical grade.

### 2.2. Preparation of Soy Drink with C. vulgaris Microalgae

For the preparation of soy drinks with microalgae, 1.5% (*w/v*) inactivated dried organic *C. vulgaris* powder was added to 50 mL of soy beverages; the concentration of the microalgae was based on the manufacturer’s daily recommendation for humans. Additionally, to sweeten the samples, 1.5% (*w/v*) xylitol was incorporated. The schematic representation is presented in Figure 1. Eight samples were prepared, four for each probiotic strain (*L. fermentum, L. rhamnosus*), according to the previous method [39]. The first sample was the control (probiotic strain and soy beverage). In the second container 1.5% *C. vulgaris* powder, probiotic strain, and soy beverage were added. The third sample, xylitol at a concentration of 1.5%, probiotic strain, and soy beverage were incorporated. In the fourth container, 1.5% xylitol, 1.5% *C. vulgaris* powder, probiotic strain, and soy beverage were added. The filling of the containers was done under sterile conditions.

### 2.3. Bacterial Cultures Preparation and Soy Drink Fermentation

*L. fermentum* and *L. rhamnosus* were activated in a sterilized MRS medium (121 °C for 15 min). Shortly, freeze-dried cells were stored at 4 °C, and activated to be viable for obtaining the inoculum. First, the bacterial cells were inoculated with 9 mL MRS broth and incubated at 37 °C for 48 h under aerobic conditions, and afterwards sub-cultured into 90 mL broth and incubated under the same conditions, according to the Pop et al. method [4]. All these procedures were performed in a sterile environment. To determine the cell concentration, the nanodrop spectrophotometer ND-1000 (ThermoFisherScientific, Massachusetts, USA) was used to determine the optical density of a cell suspension, in sterile saline solution (0.85% NaCl *w*/*v*), at a wavelength of 600 nm. The absorbance was adjusted to a McFarland concentration of 0.5, which corresponded to ~ 1.5 × 10^8^ CFU/mL and diluted in a 1:9 ratio in sterile serum to achieve a 10^7^ CFU/mL solution which was used as inoculum. The inoculum was added to each sample and represented 10% (*v*/*v*) of the sample volume. After inoculation, all the samples were incubated under aerobic condition for 24 h at 37 ^°^C, and 150 rpm in a Heidolph Rotary Incubator 1000 (Heidolph, Schwabach, Germany).

### 2.4. Determination of Cell Concentration and Probiotic Viability and pH Level

To determine the cell viability of the probiotic strains during fermentation the pour–in-plate method was used. Briefly, 1 mL of tenfold successive dilution in sterile serum was added in sterile Petri dishes, as described in previous work [40]. The process was followed by adding MRS Agar (approx. 15–20 mL) and incubating at 37 °C for 24 h.

After incubation, the colonies on each plate were counted and expressed as log10 CFU/m. pH was measured using a WTW inoLab 7110 laboratory pH meter. The cell viability and pH of the samples were measured before and after the incubation period.

### 2.5. Rheological Measurements

The viscosity of the samples was measured after fermentation with *L. fermentum* and *L. rhamnosus* as specified in Section 2.2., and soy drinks enriched with *C. vulgaris* and/or xylitol were measured with an Anton Paar MCR 72 rheometer (Anton Paar, Graz, Austria) equipped with a Peltier plate-plate system (P-PTD 200/Air). Samples were placed between the two plates, the upper one with a diameter of 50 mm, with smooth parallel plate geometry, and the lower one with a temperature control system set at 20 °C and at a distance of 1 mm [41,42]. Excess samples were removed before measurement, and samples were allowed to stand for 10 min to ensure thermal equilibrium before measurement, as reported earlier [43]. Each measurement was performed twice with a shear rate increasing linearly from 5 to 300 1/s.

### 2.6. Determination of Total Polyphenols and Antioxidant Activity from Soy Beverages with C.vulgaris and Bacteria-Folin Ciocâlteu

Total polyphenol content was investigated by measuring the absorbance at 750 nm from a primary extract complexed with the Folin-Ciocâlteu reagent. The beverage extracts were prepared as reported in previous research [44]. An amount of 1.0 mL of diluted sample (1:5) was mixed with water and filtered through membrane-filtered (0.2 μm Millipore nylon filter).

Briefly, an aliquot of the filtered sample (25 µL) was initially mixed with 1.8 mL of distilled water, followed by the addition and homogenization with 120 µL of Folin–Ciocâlteu reagent. After 5 min, 340 µL of 7.5% Na_2_CO_3_ aqueous solution was added to the mixture. The samples were put at room temperature in the dark for 90 min, and their absorbances were measured with a microplate reader (BioTek Instruments, Winooski, VT, USA). The total amount of polyphenols was expressed as mg of gallic acid equivalents (GAE)/100 g dry weight (DW) [29].

In determining antioxidant activity, compounds with anti-radical properties discolor the stable purple-red DPPH radical solution, which has a maximum absorption between 515–525 nm. The stock solution of DPPH (80 μM) was freshly prepared in 95% methanol according to a previously reported method [45]. A volume of 250 μL of DPPH solution was mixed with 35 μL of filtered sample and then was measured with the microplate reader at 515 nm absorbance. Antioxidant effectiveness was calculated using the following formula:% DPPH scavenging activity = (1 − A_s_/A_c_) × 100,(1)
where A_s_ represents the absorbance of the sample and A_c_ represents the absorbance of the control.

### 2.7. Static In Vitro Simulation of Gastrointestinal Food Digestion

The updated static in vitro digestion method developed by the INFOGEST working group was used to evaluate the viability of *L. fermentum,* and *L. rhamnosus* in the samples. The protocol extensively described by Brodkorb et al. [46] is based on sequential oral, gastric and intestinal digestion while parameters such as electrolytes, enzymes, bile, dilution, pH, and digestion time are established on available physiological data. Due to the short oral retention duration of the samples and the absence of starch in the matrix, the samples were subjected to a two-stage in vitro digestion process, mimicking the conditions of the stomach and small intestine, as previously described by Szabo et al. [47].

An aliquot of 5 mL from each type of beverage was mixed with 5 mL of simulated gastric fluid. The SGF was composed of electrolyte solutions KCl, KH_2_PO_4_, NaHCO_3_, MgCl_2_•6H_2_O, (NH_4_)_2_CO_3_, alongside a CaCl_2_(H_2_O)_2_ solution (0.03 M), porcine pepsin solution (2000 U/mL in the final digestion mixture), and water. The pH of the samples was adjusted to 3 by adding HCl (1 M), and the mixture was homogenized and incubated for 2 h in a shaking incubator New Brunswick Innova 44, Eppendorf AG, Hamburg, Germany).

For the intestinal phase, the samples were mixed with 10 mL of pre-warmed simulated intestinal fluid (SIF) to achieve a final ratio of 1:1 (*v/v*), bile extract solution, in order to reach a final concentration of 10 mM and pancreatic enzymes (100 U/mL). The pH was set to 7 using NaOH (1 M), and the mixture was homogenized and incubated at 37 °C for 2 h in a shaking incubator (95 rpm). After the process was complete, the samples were evaluated for the viability of bacteria as described in subSection 2.4.

### 2.8. Statistical Analysis

All measurements and analyses were done on three prepared samples, and the results are presented as means ± standard deviations (SD). One-way analysis of variance (ANOVA) and Tukey’s comparison test via Graph Prism Version 8.0.1. (GraphPad Software Inc., San Diego, CA, USA) and Minitab statistical software (version 16.1.0; LEAD Technologies, Inc., Charlotte, NC, USA) were applied to analyze the differences among samples with significance levels of *p* < 0.05.

## 3. Results and Discussions

### 3.1. Probiotic Viability and pH

To identify the effect of microalgae on *L. fermentum* and *L. rhamnosus* viability in fermented soy drinks, samples were kept at 4 °C and analyzed at the beginning (time 0) and at the end of the fermentation process (24 h) using the plate method (Figure 2).

At time 0, there were no significant differences in drinks (*p* > 0.05), with an average value of 6.3–7.3 log10 CFU/mL. The influence of *C. vulgaris* powder on the LAB could be observed after 24 h fermentation. At the end of fermentation, the antimicrobial effect of xylitol can be observed; the concentrations being 7.30 log10 CFU/mL in the drink inoculated with *L. fermentum* and xylitol and 7.87 log10 CFU/mL in the drink inoculated with *L. rhamnosus* and xylitol. The samples with soy drinks and xylitol decreased significantly (*p* < 0.05) for both types of bacteria, especially for *L. fermentum* and xylitol beverages compared with the control (soy drink with bacteria). In the case of *L. rhamnosus* soy beverages with xylitol, a significant decrease could be observed between the samples with supplement addition. For the control sample, no significant differences could be observed, understand the context of *L. rhamnosus*. Thus, the antimicrobial effect of xylitol depends on the species of bacteria used in the beverage. On the other hand, statistically significant differences in the viability of bacteria were identified between drinks and samples with addition of *C. vulgaris* at the end of the fermentation process. Even if initially the values were in favor of the development for *L. fermentum*-brewed drinks, after 24 h of fermentation, it could be observed that the difference in cell concentration between the two probiotic-brewed drinks was in favor of the viability for *L. rhamnosus*-brewed drink beverage. The maximum viability increase was observed in the *L. rhamnosus* samples (8.74 log10 CFU/mL) and *L. fermentum* (8.71 log10 CFU/mL) for the addition of *C. vulgaris*, followed by the samples with *C. vulgaris* and xylitol. Therefore, *C. vulgaris* powder positively affected the development of LAB, providing them with a favorable environment for growth.

These findings are sustained by the results showed by Ścieszka 2022, and these results revealed that *Lactobacillus* spp. growth medium supplemented with *C. vulgaris* at concentrations of 0.1% (*w*/*v*) and 1.5% (*w/v*), enhanced bacterial growth and shortened their phase of logarithmic growth in MRS [48]. Moreover, in another research, Ścieszka et al. [27] confirmed that the addition of *C. vulgaris* (1.5%) stimulated an increase in *Levilactobacillus brevis* (8.49 log CFU/mL) in the soybean beverages. The positive effects of microalgae on the viability of bacteria can be explained by the fact that microalgae provide nutritious and stimulating environments that help in bacterial development. Examples of such compounds include exopolysaccharides, adenine, hypoxanthine, free amino acids, and necessary vitamins and minerals [32,49]. Several authors have shown that incorporating microalgae into fermented dairy products, such as yogurt, cheese, and kefir has beneficial effects, including enhancing the concentration of LAB [49,50,51]. As the *C. vulgaris* fermented beverages included a significant number of LAB cells (8.74 log10 CFU/mL), it can be suggested that this microalga could be beneficial in the production of probiotic-rich fermented foods.

Throughout fermentation, the pH decreased, suggesting that the fermentation proceeded normally and began the production of organic acids by LAB, most notably lactic acid [28]. The findings of the study showed that the quantity of LAB in soy beverages increased after fermentation while the pH reduced. As a result of the continuing metabolic activities of the LAB toward the end of fermentation, the pH lowered to 4.5 for both types of bacteria, which can also be seen in several studies on different substrates [52,53,54]. Production of organic acids during fermentation is linked to a decrease in the pH, as was also observed in this study, while the total Titratable Acid increased as revealed by several studies [55,56]. In the present study, the concentration of organic acids was not considered relevant.

### 3.2. Rheological Measurements

The present manuscript analyzed the flow behavior of soy drink fermented with/ *L. fermentum* or *L. rhamnosus*, and with/without *C. vulgaris* enrichment and xylitol addition. The alterations in the apparent viscosity and shear stress at a temperature of 20 °C, and with constantly increasing shear rate (ẏ) from 5 to 300 1/s, are presented in Figure 3 and Figure 4a–c. As can be observed the viscosity of the samples differ in case of both LAB and also with the enrichment of the samples. Significant differences (*p* < 0.05) could be only observed between the fermented soy drink with *L. fermentum* and *L. rhamnosus.*

As can be observed from the results, the soy drink fermented with both LAB increased the viscosity of the samples from 4.55 ± 0.15 mPa·s to 8.65 ± 0.15 mPa·s with *L. fermentum* and 6.6 ± 0.5 mPa·s with *L. rhamnosus*. In contrast, the addition of *C. vulgaris* reduced the viscosity through fermentation from 4.55 ± 0.25 mPa·s to 3.75 ± 0.05 mPa·s with *L. fermentum* and increased the viscosity to 8.85 ± 0.05 mPa·s with *L. rhamnosus*. The viscosity of soybean milk began to decrease at a shear rate of 150 1/s, presenting a shear-thinning (pseudoplastic) behavior. The fermentation with *L. rhamnosus* had a higher viscosity than the samples with *L. fermentum,* which suggests superior qualities as with the increase in viscosity, the stability of the final product is better. Correlated with the obtained results at Section 3.1., pH could play an important effect on the viscosity of the samples. As the addition of xylitol and *C. vulgaris* decrease the pH, they also increase the viscosity of the samples. The fermented samples enriched with *C. vulgaris* and xylitol showed the same behavior. On the other hand, the soybean milk enriched with *L. fermentum* and xylitol (Figure 3b) presented a shear-thickening (dilatant) behavior. The consistency and viscosity of soybean milk products are important parameters, as soybean milk is constituted of a significant number of small lipid droplets diffused in water, originating from soybean seeds [43].

Although presenting a shear thickening behavior, the soy drink products presented a higher viscosity after the addition of xylitol than the unenriched soybean products, thus improving their stability. Additionally, through fermentation with *L. fermentum* or *L. rhamnosus* the indigestible carbohydrates are removed, the volatile profile and protein digestibility are enhanced, and the repellent smell of soybean-derived products is diminished [41,57]. Thus, the enrichment of soybean milk with *C. vulgaris* and fermented with these two LAB presents a functional and nutritionally improved product. Compared with similar studies soy drink after fermentation had a lower viscosity than without fermentation as presented by De et al. 2022, where the viscosity of soy milk was 17.09 ± 0.65 mPa·s [58]. This can be due to the produce lactic acid, which can influence the protein network relationship, and has an effect of the samples viscosity [59].

### 3.3. Total Polyphenols and Antioxidant Activity

For measuring the total of polyphenols and antioxidant activity, Folin-Ciocâlteu and DPPH methods were used. The results are presented in Table 1. Total polyphenols and antioxidant activity increased fermentation beverages, and the highest increases for polyphenols were observed in the samples fermented with *L. rhamnosus* and *C. vulgaris* powder compared to the control sample.

The level of total polyphenols, compared to the control sample (267.08 µg GAE/ mL), the highest content of polyphenols was found to be significantly different in samples inoculated with *L. rhamnosus* (327.26 µg GAE/ mL), respectively, (306.72 µg GAE/ mL) in samples inoculated with *L. fermentum*. Regarding the samples with xylitol and bacteria, for both strains no significant difference can be observed (*p* > 00.5). Additionally, no significant difference was found between the samples with *C. vulgaris* and xylitol.

The values obtained for antioxidant activity ranged from 301.76 ± 2.53 µM Trolox/g DW to 497.43 ± 1.28 µM Trolox/g DW. The best scavenging activity against DPPH radical was caused by the drink with *C. vulgaris*, xylitol, and *L. rhamnosus*, significantly different to the drink with *C. vulgaris*, xylitol, and *L. fermentum* (*p* < 0.05). The lowest content in antioxidant compounds was recorded by the control sample.

Previous studies have suggested that the LAB fermentation process increases the biological compounds’ bioavailability in beverages [33,60]. The obtained values are in agreement with recently published data, where the concentration of polyphenols in the beer with a 3.3 g/L addition of *C. vulgaris* was 257.81 ± 15.20 µg GAE/ mL [61]. Moreover, the fermentation process and the probiotic in which the drink is inoculated, perform an important role in total polyphenols and antioxidant activity [49,50,51,62]. Marazza et al. [63] identified two compounds, β-glucosidase, and isoflavone aglycone, as possible causes of the DPPH radical scavenging capacity of *L. rhamnosus* cultures in fermented soy drink. It is essential to mention that in this work, the beverages that had been fermented for 48 h had a high concentration of LAB, which could be responsible, along with the phenolic released from *C. vulgaris*, for the increase in antioxidant capacity. As a future perspective it will be interesting to evaluate the dynamic of the total phenolic contents and antioxidant activity during fermentation.

### 3.4. Gastric Simulation

In order to follow the bio-accessibility of the bacteria in soy beverages, the samples were subjected to the gastric and intestinal phases of the in vitro digestion protocol. Table 2 shows the pre- and post-digestion counts of LAB in soy drinks, soy drinks with *C. vulgaris*, and soy drinks with microalgae and xylitol. Results are presented in log10 CFU/mL. Before digestion, *L. rhamnosus* samples showed higher probiotic viability, but after digestion, *L. fermentum* inoculated samples showed significantly higher viability than *L. rhamnosus* samples (*p* < 0.05). During the digestion process, the bacterial count in all the samples was reduced significantly. The largest difference was evidenced in the samples with *L. rhamnosus*, especially in the samples with *C. vulgaris* and xylitol. The samples inoculated with *L. rhamnosus* were not significantly affected by the addition of *C. vulgaris* (*p* > 0.05). Therefore, no protective effect provided by *C. vulgaris* was observed in this case. The positive effects of *C. vulgaris*, can be observed in the *L. fermentum* beverages, where the soy drinks with bacteria and microalgae were significantly higher than other beverages (*p* < 0.05). The results are in agreement with previous studies [64,65]

The long-term effects of probiotics in the gut depend on them surviving the acidic environment of the digestive tract. Probiotics are degraded not only by the acidity in the stomach, but also by salts and enzymes such as pepsin and lysozyme [39,64]. Thus, it was possible to observe the impact of digestive enzymes and stomach hydrochloric acid on the bacterial stability, highlighting the viability of *L. fermentum* in acidic media.

Increasing knowledge of the possible health advantages of some algae has prompted the food industry to develop functional food products incorporating microalgae as an ingredient [27,29,34,48,52,62]. In this context, de Medeiros et al. [66] provide evidence on the prebiotic effect of *C. vulgaris* after 48 h of fermentation due to the highest glucose (16.00 g/100 g) and fructose (10.00 g/100g) concentrations. *Lactobacillus* are considered probiotics that support beneficial functions in the human body [67]. Hence, a rise in their relative abundance in the gut is desired [33]. Niccolai et al. [32] validated that microalgae with soy beverages sustain the development of *Lactobacillus* after fermentation. Ścieszka et al. highlight the use of *C. vulgaris* as natural growth stimulator of starter bacteria in fermented food production. Their results confirm that microalgae create protective circumstances for probiotics in the human digestive system [27]. Another study suggests that *L. plantarum* incorporated in soy sauce has a high survival rate through the simulated digestion process (above 6.0 log10 CFU/g).

Incorporating *C. vulgaris* into a fermented soy beverage is a novel approach that could provide nutritional and health benefits using a natural resource.

## 4. Conclusions

This research showed that the addition of 1.5% *C. vulgaris* in a fermented soy-based product positively influences the growth and development of *L. fermentum* and *L. rhamnosus*. The viability of these two LABs increased significantly after 48 h of fermentation, and the pH decreased at values between the range of 6.30–4.50. The *C. vulgaris* powder in the fermented beverage was found to lower the pH of the product to values similar to other fermented dairy products of animal origin. On the other hand, the addition of xylitol 1.5% negatively impacted the growth of probiotics. However, the soy drink with xylitol and *C. vulgaris*, because of the bacterial count, also qualifies as a fermented probiotic product. Through digestion, the low pH of the stomach and the antimicrobial action of pepsin is known to provide an effective barrier against the entry of bacteria into the intestinal tract. Regarding the viability of the probiotics after gastrointestinal simulation, the protective effect given by *C. vulgaris* powder can be observed in the case of *L. fermentum* drinks. Accordingly to the results obtained after digestion, the supportive impact of microalgae depends on the of bacteria used. Thus, enriching soy beverages with *C. vulgaris* and fermenting with these two LAB presents a functional and nutritionally improved product. Further investigation on probiotic encapsulation techniques is required to understand the components responsible for the effective distribution of bacteria through the gastrointestinal tract.

## Figures and Tables

**Figure 1 biomolecules-13-00245-f001:**
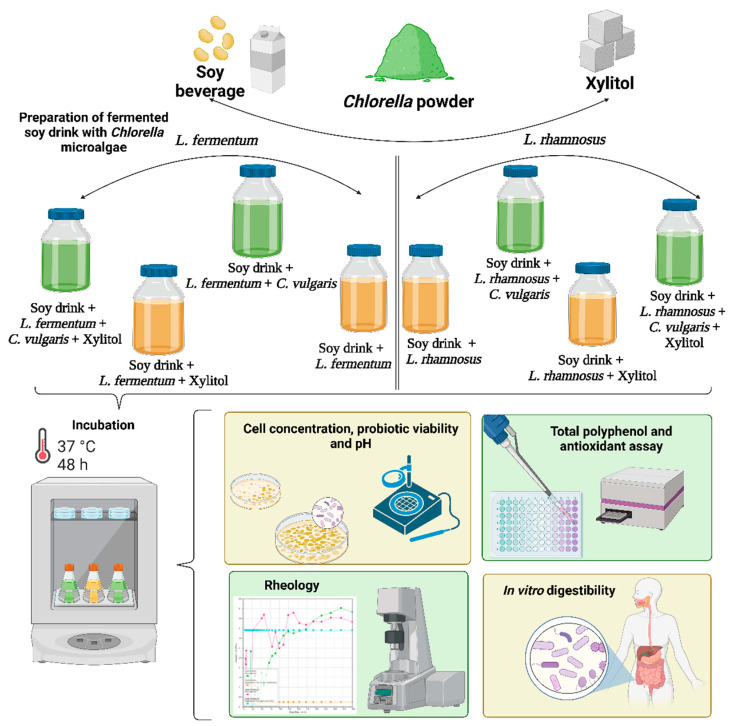
Schematic representation of the experimental design: 1. Preparation of fermented soy drink with *C. vulgaris* microalgae; 2. Determination of cell concentration; 3. Rheological measurements; 4. Determination of total polyphenols and antioxidant activity; 5. In vitro simulation of gastrointestinal food digestion.

**Figure 2 biomolecules-13-00245-f002:**
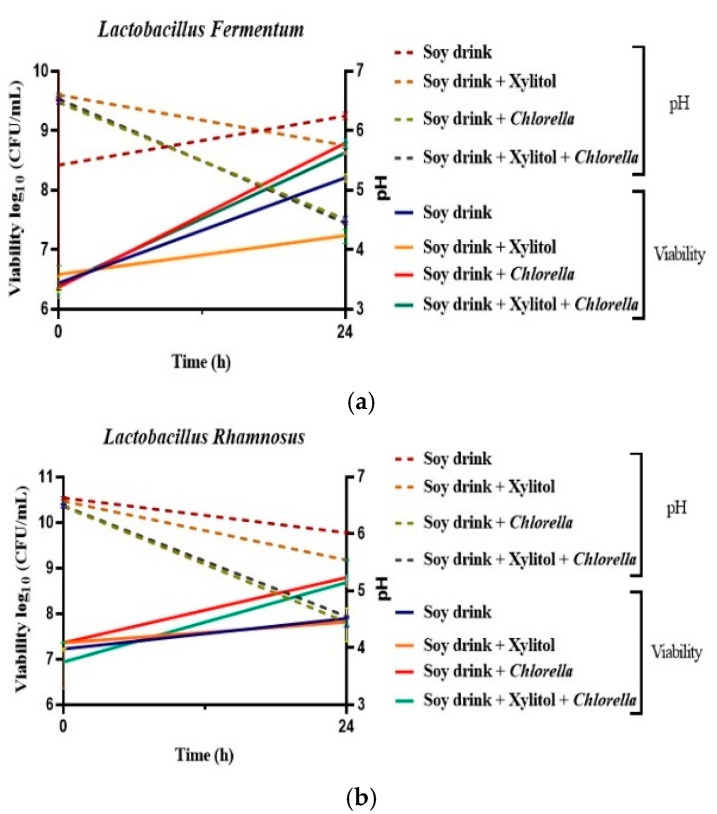
Cell viability and pH profile of the fermentation with (**a**) *L. fermentum* and (**b**) *L. rhamnosus*. Values for LAB viable cell growth and pH are displayed as mean values, log10 CFU/mL, *n* = 3, GraphPad Prism Version 8.0.1; soy drink (1); soy drink + xylitol (1.5%); soy drink + *C. vulgaris* (1.5%), soy drink + xylitol+ *C. vulgaris* (1.5%); CFU/mL (colony-forming units/milliliter of the sample).

**Figure 3 biomolecules-13-00245-f003:**
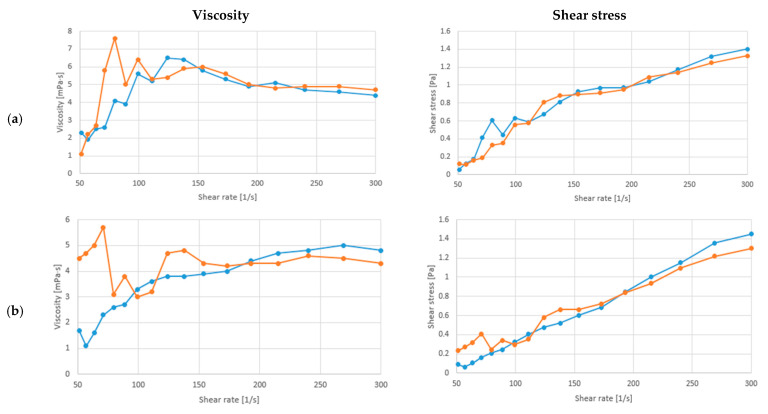
Relationship between viscosity/shear stress and shear rate at 20 °C of the samples with (**a**) Soy drink + *L. fermentum,* (**b**) Soy drink + *L. fermentum +* Xylitol, (**c**) Soy drink *+ L. fermentum* + *C. vulgaris +* Xylitol (experiments effectuated in duplicate red and blue line).

**Figure 4 biomolecules-13-00245-f004:**
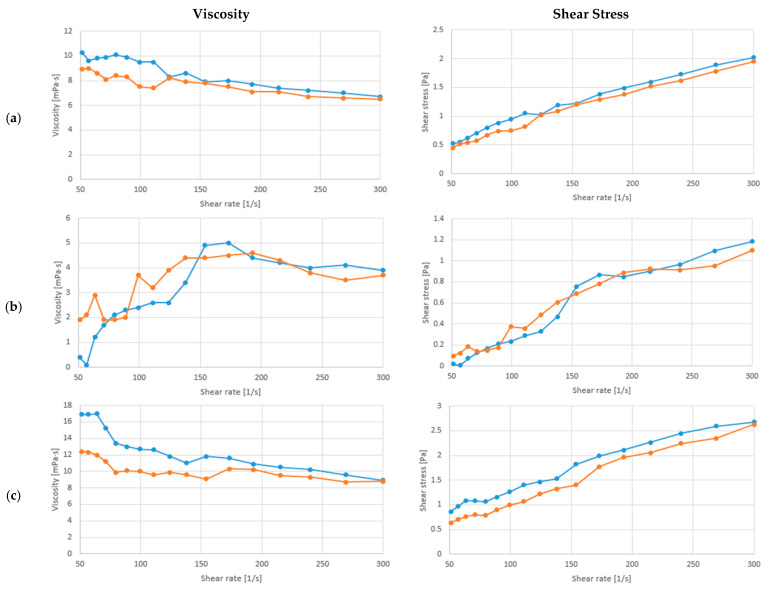
Relationship between viscosity/shear stress and increasing shear rate at 20 °C of the sample with (**a**) Soy drink + *L. rhamnosus,* (**b**) Soy drink + *L. rhamnosus +* Xylitol, (**c**) Soy drink *+ L. rhamnosus* + *C. vulgaris +* Xylitol (experiments effectuated in duplicate red and blue line).

**Table 1 biomolecules-13-00245-t001:** Total phenolic content (TPC) and antioxidant activity (DPPH) of the beverage samples.

Beverages	TPC (µg GAE/mL)	DPPH (µM Trolox/g DW)
Soybean drink with *C. vulgaris*	267.08 ±1.51 ^C^	301.76 ± 2.53 ^E^
Soy drink with *C. vulgaris* and xylitol	263.28 ± 1.31 ^C^	347.44 ± 1.17 ^D^
Soy drink with *C. vulgaris* and *L. rhamnosus*	327.26 ± 0.31 ^A^	450.13 ± 1.23 ^B^
Soy drink with *C. vulgaris* and *L. fermentum*	306.72 ± 1.33 ^B^	422.56 ± 0.65 ^C^
Soy drink with *C. vulgaris*, xylitol, and *L. rhamnosus*	321.40 ± 1.63 ^A^	497.43 ± 1.28 ^A^
Soy drink with *C. vulgaris*, xylitol, and *L. fermentum*	304.42 ± 1.72 ^B^	453.29 ± 1.21 ^B^

Values are expressed as mean ± standard deviation. For each characteristic, identically superscript capital letters indicate no significant differences (*p* > 0.05) between samples.

**Table 2 biomolecules-13-00245-t002:** Viable counts (log10 CFU/mL) of probiotic bacteria in different treatments during storage time.

Samples	Before Digestion (log10 CFU/mL)	After Digestion (log10 CFU/mL)
Soy drink + *L. fermentum*	8.20 ± 0.60 ^b,A^	6.67 ± 0.64 ^b,B^
Soy drink + *L. fermentum* + Xylitol	7.30 ± 0.12 ^c,A^	6.62 ± 0.53 ^b,B^
Soy drink + *L. fermentum* + *C. vulgaris*	8.71 ± 0.64 ^a,A^	6.81 ± 0.45 ^a,B^
Soy drink + *L. fermentum* + Xylitol + *C. vulgaris*	8.62 ± 0.21 ^a,A^	6.72 ± 1.19 ^b,B^
Soy drink + *L. rhamnosus*	7.71 ± 0.44 ^b,A^	5.67 ± 0.69 ^c,B^
Soy drink + *L. rhamnosus* + Xylitol	7.87 ± 0.62 ^b,A^	5.54 ± 1.34 ^c,B^
Soy drink + *L. rhamnosus* + *C. vulgaris*	8.74 ± 0.49 ^a,A^	5.58 ± 0.58 ^c,B^
Soy drink + *L. rhamnosus* + Xylitol + *C. vulgaris*	8.68 ± 0.80 ^a,A^	5.51 ± 0.54 ^c,B^

Values are expressed as mean ± standard deviation. ^A,B^—the statistical differences between viable counts of bacteria before and after digestion; ^a,b,c^—the statistical differences between viable counts of bacteria in one testing sample. For each characteristic, identically superscript letters indicate no significant differences (*p* > 0.05) between samples.

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
