# Peer review of "Development of a Fermented Beverage with Chlorella vulgaris Powder on Soybean-Based Fermented Beverage"

_biomolecules, 2023, doi:10.3390/biom13020245_

Round 1

Reviewer 1 Report

Reviewer Comments to Author 

Brief abstract 

In this work, a new soy-based fermented beverage has been obtained using Chlorella vulgaris as a substrate for the development of two lactic acid bacteria. This is an interesting and quite innovative idea that addresses a useful topic to produce new beverages with functional properties. However, a general and in-depth revision of the manuscript is needed; most of the sections should be revised to improve it. In particular, the material and methods section needs a major restructuring. Also, the results and discussion should be improved by analysing the results and including new and recent references related to this topic. Finally, the conclusions should be revised to ensure that they are supported by the data. For all these reasons, I consider that it is unacceptable and should be rewritten and include a few more experiments.

 Below there are some comments to improve the article: 

General comments  

- Check the spelling in the full text. There are several errors in the names of micro-organisms: C. vulgaris.  

 - In general, the material and methods are confusing and incomplete. Please review carefully and clarify all aspects to improve readability 

 - I would recommend revising the English of the entire text. 

Specific comments 

Abstract 

  • Line 22: In "During fermentatio, a bacterial concentration...", the authors should indicate which bacteria were found.  

  •  Lines 23-24: The authors state that "The polyphenol content and antioxidant capacity of the diet were significantly improved after fermentation". However, this statement is not supported by the data obtained in the study. 

Introduction 

  • Lines 55 and 57: please match the citation number with the authors 

  •  Line 60: please check if article 36 is related to the information indicated in this sentence.  

  •  Lines 67-69: The objective is not clear enough and is not representative of the study conducted. Please modify it to improve it. 

Material and Methods 

  • Lines 72-81: This section is quite confusing. Please rewrite to clarify. This section includes different materials: microorganisms, reagents, etc., without clear separation.  

  • Line 76: There is no information on the commercial soy beverage used in the study. A physico-chemical and microbial characterisation of the product would be necessary. It would be relevant to know the nutrient concentrations and whether this product is sterile.  

  • Line 76: Please provide more information on Chlorella powder; is it an inactivated product; its exact composition? A physico-chemical and microbial characterisation of the product would be necessary to know the nutrient concentrations and whether this product is sterile.  

  • Sections 2.2 and 2.3. Please rewrite these sections to adequately explain the fermentation process...: 

  • Line 82: The title is not appropriate to the content of the section. Please rewrite it.  

  • Lines 83-91: This section is also very incomplete and confusing. It would be helpful to give a full explanation of the process followed to obtain the final product. 

  • Some ideas that are not included in this section are:  

  • How the soya beverage was prepared prior to inoculation.  

  • Inoculum concentration used  

  • Why is xylitol used? This is not justified in the text.  

  • How many replicates were made?  

  • How much soya beverage was used 

  • What were the fermentation conditions: oxygen, agitation...?  

  •   

  • Figure 1: This figure does not include relevant information about the process. Moreover, its caption is not related to the figure itself. Please modify the figure and/or the caption.  

  • Line 97: The title is not representative of the content of the section. Please rewrite this section.  

  • Line 98: Please indicate here the storage conditions of the bacteria and why they were activated.  

  • Line 107: What is the composition of the sterile serum?  

  • Lines 116-118: Please put this sentence in another paragraph.  

  • Line 120: It would be appropriate to include the samples before fermentation in the experimental design.  

  • Line 126: More information on the parameters used in the rheological analysis should be included in this section.  

  • Line 162: The authors assume that the soy beverage is starch-free. However, it is a soy beverage and could contain starch. A characterisation of the beverage is needed. 

  •  

Results and discussion 

  • Line 189: This title is not appropriate for the content of the section. Please amend it.  

  • Lines 191-193: This paragraph is part of the discussion. Please explain the results first  

  • Line 195: Revise the temperature used during fermentation and the time. Figure 2: Please modify the names of the microorganisms; they are not spelled correctly. Include error bars in the figure. The figure legend is too long. Please shorten it.  

  • Line 207: Why was xylitol used? Please justify this and the literature used for this. Is the antimicrobial effect of xylitol based on statistical differences?  

  • Line 212-213: Please confirm if you observe differences between beverages with xylitol and without added supplement. In the case of L. rhamnosus this is not clear.  

  • Lines 218-219: Data are needed on the effect of Chlorella on LAB growth on synthetic media.  

  • Line 231: Some data are needed on the effect of Chlorella on LAB growth on synthetic media.  

  • Analysis of titratable acidity would be relevant to determine the concentration of acids produced.  

  •  
  • Section 3.2 This section is rather confusing. Data on growth on xylitol medium is missing. Please include information on the flow parameter in Material and Methods.  

  • Figure 3 is not very clear. The information on the lines, and the numbers and axis text are not readable.  

  • Line 245: Fermentations e and f are not included in the experimental design in Material and Methods.  

  • Lines 250-251: Please explain the differences in behaviour between L. fermentum and L. rhamnosus.  

  • Lines 263-264: There is no data to support this argument.  

  •  
  • Section 3.3. This section should be revised in depth. First, explain the table and the results shown in it 

  • Lines 277-279: Please put this paragraph into the discussion.  

  • Table 1 is incomplete, why is there no data for fermentations with xylitol? The data in the table are from the end of fermentation; it would be useful to include data from the beginning of fermentation as well. Without these data it is not possible to conclude what the effect of chlorella on phenolic and antioxidant activity is.  

  •  
  • Section 3.4: The discussion in this section is insufficient. There is no explanation of the effect of digestion and comparison with other authors. Table 2: There was no difference between fermentations with the same LAB: xylitol, and Chlorella did not affect bacterial concentration.  Data before digestion are not the same as in section 3.1. Please compare statistically the samples before and after digestion.  

  • Lines 330-337: Also, this paragraph would be better as part of the discussion in the first section.  

Conclusions:  

  • The effect of xylitol is not so clear; please revise and justify it better.  

  • Lines 349-351: This sentence does not seem appropriate for inclusion in the conclusions because it is not part of the study.  

  • Lines 356-357: No protective effect of C. vulgaris potency to probiotics seems to be observed. Please justify this.

Author Response

Dear reviewer,

We want to express our appreciation for your valuable comments, which improve the quality of the paper.

The corrections are marked in the manuscript with Track Changes.

Our revised sections, according to your comments, are listed as follows.

Comment:

Response:

1

- Check the spelling in the full text. There are several errors in the names of micro-organisms: C. vulgaris.  

 - In general, the material and methods are confusing and incomplete. Please review carefully and clarify all aspects to improve readability.   

 - I would recommend revising the English of the entire text. 

We highly appreciate your consideration of the manuscript. Your comments and suggestions helped us improve the flow of the paper and scientific sound. Please find below the response to each comment and suggestion.

The manuscript has been carefully reviewed by an experienced editor whose first language is English.

2

Abstract 

·         Line 22: In "During fermentatio, a bacterial concentration...", the authors should indicate which bacteria were found.  

According to the reviewer, information was added to the manuscript.

·          Lines 23-24: The authors state that "The polyphenol content and antioxidant capacity of the diet were significantly improved after fermentation". However, this statement is not supported by the data obtained in the study.

The information from the paragraph has been modified.

3

Introduction 

·         Lines 55 and 57: please match the citation number with the authors.   

The references have been matched, and additionally, we have separated the references.

·         Line 60: please check if article 36 is related to the information indicated in this sentence.

Article 36 has been checked, and we have rewritten the sentence.

Now line 60: "whereas C. vulgaris (1%, 2%, and 3%) enhanced firmness and springiness, but decreased cohesiveness and meltability."

·         Lines 67-69: The objective is not clear enough and is not representative of the study conducted. Please modify it to improve it. 

Thanks for your feedback and recommendation. Now lines 67-69: "To better understand the action of biological compounds from C. vulgaris powder, the present study was conducted to investigate the effects of the addition of microalgae on microbiological and biochemical characteristics of probiotic soybean beverages for possible use as a functional and lactose-free product."

4

Material and Methods 

·         Lines 72-81: This section is quite confusing. Please rewrite to clarify. This section includes different materials: microorganisms, reagents, etc., without clear separation.  

The information from the paragraph has been modified.

·         Line 76: There is no information on the commercial soy beverage used in the study. A physico-chemical and microbial characterisation of the product would be necessary. It would be relevant to know the nutrient concentrations and whether this product is sterile.  

The information has been added in section 2.1, as suggested by the reviewer.

·         Line 76: Please provide more information on Chlorella powder; is it an inactivated product; its exact composition? A physico-chemical and microbial characterisation of the product would be necessary to know the nutrient concentrations and whether this product is sterile.  

The information has been added in section 2.1, as suggested by the reviewer.

·         Sections 2.2 and 2.3. Please rewrite these sections to adequately explain the fermentation process...: 

Sections 2.2 and 2.3 has been modified, as suggested by the worthy reviewer.

·         Line 82: The title is not appropriate to the content of the section. Please rewrite it.

As suggested by the reviewer, we have agreed. For it, we proposed a more suggestive title: "Preparation of soy drink with C. vulgaris microalgae"

·         Lines 83-91: This section is also very incomplete and confusing. It would be helpful to give a full explanation of the process followed to obtain the final product.

The information from the paragraph has been modified.

Some ideas that are not included in this section are:

How the soya beverage was prepared prior to inoculation.  

The information has been added: "For the preparation of soy drink with microalgae, 1.5% (w/v) inactivated dried organic C. vulgaris powder was added in 50 ml of soy beverages;"

·         Inoculum concentration used  

The inoculum concentration was specified: "a 107 CFU/ml solution which was used as inoculum."

·         Why is xylitol used? This is not justified in the text.  

The information has been justified: "Also, to sweeten the samples, 1.5% (w/v) xylitol was incorporated."

·         How much soya beverage was used?   

The information has been added: "For the prepation of soy drink with microalgae, 1.5% (w/v) inactivated dried organic C. vulgaris powder was added in 50 ml of soy beverages;"

·         What were the fermentation conditions: oxygen, agitation...?  

The fermentation conditions were specified: "After inoculation, all the samples were incubated under aerobic condition for 24 h at 37 °C and 150 rpm in a Heidolph Rotary Incubator 1000 (Heidolph, Schwabach, Germany)."

·         Figure 1: This figure does not include relevant information about the process. Moreover, its caption is not related to the figure itself. Please modify the figure and/or the caption.  

As suggested by the reviewer, we have agreed, which is why we improved it.

·         Line 97: The title is not representative of the content of the section. Please rewrite this section.  

As suggested by the reviewer, we have agreed. For it, we proposed a more suggestive title: "Bacterial cultures preparation and soy drink fermentation".

·         Line 98: Please indicate here the storage conditions of the bacteria and why they were activated.  

The information has been added: "Shortly, freeze-dried cells were stored at 4 °C and activated to be viable for obtaining the inoculum."

·         Line 107: What is the composition of the sterile serum?  

The information has been added: "sterile saline solution (0.85% NaCl w/v)".

·         Lines 116-118: Please put this sentence in another paragraph.  

The paragraph has been moved.

·         Line 120: It would be appropriate to include the samples before fermentation in the experimental design.  

Thank you for the recommendation. As suggested, these samples have been included in the experimental design and also in the description we have modified accordingly: "The viscosity of the samples after fermentation with L. fermentum and L. rhamnosus" as specified in section 2.2., "and also soy-drink enriched with C. vulgaris and/or xylitol were measured with an Anton Paar MCR 72 rheometer (Anton Paar, Graz, Austria) equipped with a Peltier plate-plate system (P-PTD 200/Air)."

·         Line 126: More information on the parameters used in the rheological analysis should be included in this section.  

Kindly thank you for this suggestion. In the revised version of the manuscript, we included more information on the used parameters: "Samples were placed between the two plates, the upper one with a diameter of 50 mm (PP-50-67300), with smooth parallel plate geometry, and the lower one with a temperature control system set at 20 °C, and at a distance of 1 mm [1,2]. Excess samples were removed before measurement, and samples were allowed to stand for 10 minutes to ensure thermal equilibrium before measurement. Each measurement was performed twice with a shear rate increasing linearly from 5 to 300 1/s. To analyze the acquired results from the rheological measurements, the RheoCompassTM software was employed, with the results reported as ± SD."

·         Line 162: The authors assume that the soy beverage is starch-free. However, it is a soy beverage and could contain starch. A characterisation of the beverage is needed. 

According to the reviewer, a characterization of the beverage was added to the manuscript to highlight the slight quantity of carbohydrates in soy beverages, which correspond to less than 1% starch content, according to the literature.

Choct, M., et al., Soy Oligosaccharides and Soluble Non-starch Polysaccharides: A Review of Digestion, Nutritive and Anti-nutritive Effects in Pigs and Poultry. Asian-Australasian Journal of Animal Sciences, 2010. 23(10): p. 1386-1398.

5

Results and discussion 

·         Line 189: This title is not appropriate for the content of the section. Please amend it.  

As suggested by the reviewer, we have agreed. For it, we proposed a more suggestive title: "Evolution of probiotic bacterial strains during fermentation".

·         Lines 191-193: This paragraph is part of the discussion. Please explain the results first.  

The paragraph has been removed.

·         Line 195: Revise the temperature used during fermentation and the time. Figure 2: Please modify the names of the microorganisms; they are not spelled correctly. Include error bars in the figure. The figure legend is too long. Please shorten it.  

As suggested by the reviewer, the correction has been made throughout figure 2; the legend has been rewritten.

·         Line 207: Why was xylitol used? Please justify this and the literature used for this. Is the antimicrobial effect of xylitol based on statistical differences?  

As suggested by the reviewer, the justification of the xylitol use has been made in the text. Also, we have rewritten the effect of xylitol based on statistical differences to be more comprehensive.

·         Line 212-213: Please confirm if you observe differences between beverages with xylitol and without added supplement. In the case of L. rhamnosus this is not clear.  

The suggestions of worthy reviewer have been implemented in the text.

Now: "In the case of L. rhamnosus soy beverages with xylitol, a significant decrease can be observed between the samples with supplement addition and without supplement. For the control sample, no significant differences can be observed, understand the context of L. rhamnosus."

·         Lines 218-219: Data are needed on the effect of Chlorella on LAB growth on synthetic media.  

As the reviewer suggests, data regarding the influence of the algae on the lactobacillus growth in MRS media was added. Please see lines 272-275.

·         Line 231: Some data are needed on the effect of Chlorella on LAB growth on synthetic media.  

Data was added in lines 272-275.

·         Analysis of titratable acidity would be relevant to determine the concentration of acids produced.  

Indeed, the titratable acidity will provide some information regarding the produced acids amount. The decrease in the pH is correlated to the accumulation of some organic acids, increasing titratable acidity. Information was added as a correlation to the scientific literature in line 293.

Section 3.2

·         This section is rather confusing. Data on growth on xylitol medium is missing. Please include information on the flow parameter in Material and Methods.  

Kindly thank you for raising this valuable comment. Within the present revision, we corrected the results and discussion section and integrated the effect of xylitol on the experiments.

·         Figure 3 is not very clear. The information on the lines, and the numbers and axis text are not readable.  

Thank you very much for this valuable comment. As you recommended, every figure has been remade, and now the numbers and axis are more readable.

·         Line 245: Fermentations e and f are not included in the experimental design in Material and Methods.  

Thank you for every valuable comment. We revised the whole manuscript and corrected the specified sections.

·         Lines 250-251: Please explain the differences in behaviour between L. fermentum and L. rhamnosus.  

Thank you for the correction. The difference between the fermentation with these two LAB has been highlighted.

·         Lines 263-264: There is no data to support this argument.  

Kindly thank you for the observation; we included supporting data for every statement.

Section 3.3.

·         This section should be revised in depth. First, explain the table and the results shown in it.   

As suggested by the reviewer, we have agreed. For it, the discussion part has been improved.

·         Lines 277-279: Please put this paragraph into the discussion.

The paragraph has been removed.

·         Table 1 is incomplete, why is there no data for fermentations with xylitol? The data in the table are from the end of fermentation; it would be useful to include data from the beginning of fermentation as well. Without these data it is not possible to conclude what the effect of chlorella on phenolic and antioxidant activity is.

The authors have added the data for the fermentation with xylitol.

The data from the beginning of the fermentation was not evaluated, as the final product characterization regarding the antioxidant and total polyphenol activity was the purpose of the analysis. A statement was added in line 407.

Section 3.4:

·         The discussion in this section is insufficient. There is no explanation of the effect of digestion and comparison with other authors. Table 2: There was no difference between fermentations with the same LAB: xylitol, and Chlorella did not affect bacterial concentration.  Data before digestion are not the same as in section 3.1. Please compare statistically the samples before and after digestion.  

As suggested by the reviewer, we have agreed. For it, the discussion part has been improved by comparing the results with other authors. Also, the data and statistic part have been modified.

·         Lines 330-337: Also, this paragraph would be better as part of the discussion in the first section.  

The paragraph has been moved to the first section.

Conclusions:  

·         The effect of xylitol is not so clear; please revise and justify it better.  

The information has been justified, and the conclusion part has been rewritten.

·         Lines 349-351: This sentence does not seem appropriate for inclusion in the conclusions because it is not part of the study

As suggested by the reviewer, the sentence has been removed.

·         Lines 356-357: No protective effect of C. vulgaris potency to probiotics seems to be observed. Please justify this.

The conclusion has been rewritten to be more comprehensive with the protective effect of C. vulgaris.

Reviewer 2 Report

The manuscript (biomolecules-2108672) reported that the suitability of C. vulgaris biomass as a substrate for Lactobacillus fermentum and Lactobacillu rhamnosus development and fermentation in vegetal soybean beverages and to evaluate the fermented product in terms of bacterial viability, antioxidant capacity, and in vitro bioaccessibility. The study is interesting to food industry. Functional beverages made from plant-based is important to human health, therefore, we should pay much more attention to it. However, the safety of Chlorella vulgaris to fermentation microorganism and human body needs to be pointed out in the article. Others, the data need to be analyzed for significance.

Author Response

Dear reviewer,

Please consider our appreciation for your time and effort in the review process. Your statements and suggestions are valuable for us.

The corrections are marked in the manuscript with Track Changes.

Comment:

Response:

1

The manuscript (biomolecules-2108672) reported that the suitability of C. vulgaris biomass as a substrate for Lactobacillus fermentum and Lactobacillus rhamnosus development and fermentation in vegetal soybean beverages and to evaluate the fermented product in terms of bacterial viability, antioxidant capacity, and in vitro bioaccessibility. The study is interesting to food industry. Functional beverages made from plant-based is important to human health, therefore, we should pay much more attention to it. However, the safety of Chlorella vulgaris to fermentation microorganism and human body needs to be pointed out in the article. Others, the data need to be analyzed for significance.

Thank you for the valuable comments. Based on these, the quality of the paper was improved.

Information has been added in the manuscript about the algae’s impact on human health.

"Before releasing a new microalgal-based food for dietary purposes, it must be shown that the ingredient is safe. For it, de Mello-Sampayo et al. [30] evaluated carotenogenic biomass (orange) resulting from induced stresses of C. vulgaris. The consumption of C. vulgaris did not reveal any indicators of toxicity at dosages surpassing the suggested human carotenoid intake level. These findings point to the potential of microalgae as a source of carotenoids, suggesting it may have health advantages if consumed by humans."

Also, all the data were analyzed for significance.

Round 2

Reviewer 1 Report

Brief abstract 

The manuscript has been improved by the authors. They have followed most of the Reviewers' comments, and they have properly justified their responses. However, there are still some errors in the manuscript. Below there are some comments to improve the article: 

  • - Please check spelling names of microorganisms in the full text. There are some errors.

  • - Please check the citation number with the authors.    

  • - Figure 2: Analysis of the data showed that, in the case of L. rhamnosus, no differences were observed between beverages with xylitol (soy beverage+Xylitol) and without added supplement (soy beverage). In contrast, in the case of L. fermentum there are differences between the soy drink and the soy+xylitol drink. Thus, the antimicrobial effect of xylitol is not so clear and depends on the strain and/or species of bacteria used. Please clarify this aspect.  

  • - Figures 3 and 4: Please indicate the meaning of the colour of the lines 

  •  Gastric simulation: No statistical differences were observed between the L. rhamnosus fermentations after the digestion process. Therefore, no protective effect provided by Chlorella is observed in this case   

  • - Please review the conclusions based on the above comments.

Author Response

Dear reviewer,

We are grateful for your valuable comments, which improve the quality of the paper.

The corrections are marked in the manuscript with Track Changes.

Our revised sections, according to your comments, are listed as follows.

Comment:

Response:

The manuscript has been improved by the authors. They have followed most of the Reviewers' comments, and they have properly justified their responses. However, there are still some errors in the manuscript. Below there are some comments to improve the article: 

We highly appreciate your consideration of the manuscript. Your comments and suggestions helped us improve the flow of the paper and scientific sound. Please find below the response to each comment and suggestion.

Please check spelling names of microorganisms in the full text. There are some errors.

The name of the microorganisms has been checked and modified.

Please check the citation number with the authors. 

As suggested by the reviewer, the citation number has been checked and modified.

Figure 2: Analysis of the data showed that, in the case of L. rhamnosus, no differences were observed between beverages with xylitol (soy beverage+Xylitol) and without added supplement (soy beverage). In contrast, in the case of L. fermentum there are differences between the soy drink and the soy+xylitol drink. Thus, the antimicrobial effect of xylitol is not so clear and depends on the strain and/or species of bacteria used. Please clarify this aspect.  

The information from the paragraph has been modified.

Figures 3 and 4: Please indicate the meaning of the colour of the lines.   

Thank you for this comment. We have included the meaning of the colors: "(experiments effectuated in duplicate red and blue line)"

Gastric simulation: No statistical differences were observed between the L. rhamnosus fermentations after the digestion process. Therefore, no protective effect provided by Chlorella is observed in this case

According to the reviewer, information was added to the manuscript.